# A deep tensor-based approach for automatic depression recognition from speech utterances

Sandeep Kumar Pandey[1]☺*, Hanumant Singh Shekhawat[1]☺, S. R. M. Prasanna[2‡], Shalendar Bhasin[3‡], Ravi Jasuja[3,4‡]

1 Electronics and Electrical Engineering Dept, Indian Institute of Technology Guwahati, Assam, India, 2 Electrical Engineering Dept, Indian Institute of Technology Dharwad, Dharwad, Karnataka, India, 3 Brigham and Womens Hospital, Harvard Medical School, Boston, MA, United States of America, 4 Function promoting Therapies, Waltham, MA, United States of America

☺ These authors contributed equally to this work.
‡ SRMP, SB and RJ also contributed equally to this work and share senior authorship to this work.
* sandeep.pandey@iitg.ac.in

## Abstract

Depression is one of the significant mental health issues affecting all age groups globally. While it has been widely recognized to be one of the major disease burdens in populations, complexities in definitive diagnosis present a major challenge. Usually, trained psychologists utilize conventional methods including individualized interview assessment and manually administered PHQ-8 scoring. However, heterogeneity in symptomatic presentations, which span somatic to affective complaints, impart substantial subjectivity in its diagnosis. Diagnostic accuracy is further compounded by the cross-sectional nature of sporadic assessment methods during physician-office visits, especially since depressive symptoms/severity may evolve over time. With widespread acceptance of smart wearable devices and smartphones, passive monitoring of depression traits using behavioral signals such as speech presents a unique opportunity as companion diagnostics to assist the trained clinicians in objective assessment over time. Therefore, we propose a framework for automated depression classification leveraging alterations in speech patterns in the well documented and extensively studied DAIC-WOZ depression dataset. This novel tensor-based approach requires a substantially simpler implementation architecture and extracts discriminative features for depression recognition with high f1 score and accuracy. We posit that such algorithms, which use significantly less compute load would allow effective onboard deployment in wearables for improve diagnostics accuracy and real-time monitoring of depressive disorders.

## 1 Introduction

Depression is a mental health issue often characterized by low mood, sadness, and negative thoughts, loss of interest in day-to-day activities, and is often associated with an individual's

**Data Availability Statement:** The data underlying the results presented in the study are available from https://dcapswoz.ict.usc.edu/.

**Funding:** This research was supported under the India-Korea joint program cooperation of science and technology by the National Research Foundation (NRF) Korea (2020K1A3A1A68093469), the Ministry of Science and ICT (MSIT) Korea and by the Department of Biotechnology (India) (DBT/IC-12031(22)-ICD-DBT).

**Competing interests:** The authors declare that they have no competing interests.

inability to cope up with stressful events [1]. According to a report by W.H.O., clinical depression is one of the primary causes of disability [2]. Also termed as Major Depressive Disorder (MDD), depression increases an individual's risk of suicide ideation [3]. Several studies in the recent years have shown that people who commit suicide often meet the criteria for clinical diagnoses of depressive illness [4, 5]. Depression is among the most treatable of mental disorders. Between 80% and 90% of people with depression eventually respond well to treatment. Almost all patients gain some relief from their symptoms. However, definitive diagnosis in sporadic visits to the treating psychologists presents a challenge since MDD presentation evolves over time and a cross-sectional assessment alone has limited diagnostic accuracy. Accordingly, diagnostic frameworks, which could passively assist in the diagnoses and management, of clinical depression present a substantial unmet need. Current standard of care for the diagnoses of clinical depression involves clinical interviews by psychologists and administration of the standard Hamilton Rating Scale, PHQ-8 rating system to classify symptomatic presentation through a depression score for the individual [6, 7]. However, this method is subjective and time-consuming. These extant methods rely primarily on the self-report measures during interviews when the depressive behavior has been manifested. By design, the prevailing methods are not amenable to proactive, unobtrusive monitoring to prevent an individual's progression into depressed state. Additionally, the reliance on a psychologist's ability to deem someone as depressed or not is susceptible to individual clinician's appraisal bias. Non-intrusive monitoring through wearables and embedded classification algorithms presents an exciting opportunity to mitigate clinician subjective bias and provide a proactive, companion diagnostic framework. These longitudinal assessments can also be effectively integrated with various serum biomarkers such as lower serotonin levels [8], impaired functioning of neurotransmitter gamma-amino butyric acid (GABA), etc., which have been shown to be strong correlates of mental health-related indications [9]. However, these invasive biomarkers are not frequently monitored prior to explicit evidence of depressive disorder. We posit that depression progression or recognition in individuals has to be proactive and multifactorial such that subjectivity in physician's assessment can be reduced through high fidelity, data-driven algorithmic insights. Several research groups have begun to make headway in studies involving speech signal-based depression recognition [1], eye movements [10], facial activity [11], gesturing [12], slumped posture [13], etc. These markers help in automatic diagnoses of alterations in the depressive states without intruding into the patient's activities of daily living. They can be employed in wearable smart devices such as smartwatches, smartphones, etc., to continuously monitor the individual's mental state.

Depression recognition from behavioural signals such as speech, facial expressions, etc., has fostered interdisciplinary effort from research teams due to its challenging and complex physiological presentation. Several studies have pursued feature extraction and learning strategies for depression recognition from speech. For instance, the work in Alghowinem et al. [14] investigated the effect of segment level as well as prosodic features on the classification of depressed speech from normal controls. The authors pointed out that statistical functionals computed from low-level features lose information resulting in inferior performance than segment-level features. Interestingly, the studies performed by Alghowinem et al. [15] explored speech style as an aspect of depressed vs. normal speech with gender classification as a precursor to improving the recognition performance. It was found that several speech features such as MFCC, intensity, and energy features were of significance when both male and female participant's speech was considered. However, shimmer and RMS energy features were of prominence for female only depression classification, and voice quality was a stratifying marker for the male participants only. An investigation on temporal features revealed that the response time and average syllable duration were longer in depressed subjects. In contrast, the

interaction involvement and articulation rate were higher in healthy controls. Another interesting study reported by Long et al. [16] examined several speech types such as read speech, interviews, and picture description and emotion types such as positive, negative, and neutral for their discriminative power for depression versus normal speech classification. Experiments on a dataset of 74 subjects using an SVM classifier demonstrated that interview speech and neutral emotion contribute more towards recognition of depression from speech than other speech and emotion types. The study in [17] introduced a new dataset PRIORI, collected from everyday smartphone conversation recordings and utilized it to study the change of emotional activation and valence in depressed and manic phases of Bipolar Disorder. Furthermore, in an independent research study, Cummins et al. [18] investigated the effect of speaker normalization for depression classification performance as mental-health disorders are highly speaker-specific, and also, the speakers for depressed and healthy controls were different. Feature normalization for reducing speaker variabilities were shown to improve recognition performance when MFCC and formant-based features were used. All these techniques relied on hand-crafted features and traditional classifiers such as Gaussian Mixture Models (G.M.M), Support Vector Machines (SVM), etc., focusing on identifying relevant feature set for robust classification of depressed speech from healthy controls.

Multimodal approaches using audio, text, and facial geometry features have also been investigated [19–23]. Alghowinem et al. investigated the fusion of information from speech, head pose, and eye gaze behaviors for depression/normal classification on a dataset of 30 depressed and 30 healthy controls collected by Black Dog Institute [19, 24]. The authors leveraged different feature selection and fusion techniques, and found that t-test based feature selection performed well for binary depression/normal classification. Moreover, the individual modality's performance was also reported, with speech showing the maximum recognition accuracy of 83%, further strengthening the idea that speech alone contains sufficient information for robust depression recognition. Also, in [20], new video and text features are proposed, and a hybrid of deep and shallow networks are used for depression classification using audio, video, and text modalities. Individual modalities such as audio and video were modelled using DCNN-DNN based system, while text modality was modelled using Paragraph Vector (P.V.) based SVM system. Moreover, in [22], an LSTM based system was explored to simultaneously model depression from audio and text sequences without performing explicit topic modelling of the content of the interviews. Also addressing the AVEC 2016 depression sub challenge, the work in [23] used i-vector framework with MFCC features for audio data modelling and geometrical features along with polynomial parametrization of facial landmarks was used in a late-fusion fashion for depression classification. From recent literature in depression classification, it is prominent that different combinations of modalities have been explored to demonstrate a robust system. However, another major observation which can be derived from such studies is the higher performance using audio modality, which serves as a motivating factor to further explore audio based depression recognition.

With progress in the deep learning field and increased computation efficiency, the dependence on hand-crafted features is reduced. Deep learning has facilitated efficient end-to-end modelling of complex paralinguistic phenomenon which is difficult to assess using traditional techniques. Deep learning has been successfully applied to the task of automated diagnosis and modelling such as Bipolar Disorder [17], anxiety [25], alzheimer's dementia [26], clinical depression [27] etc. Much of the recent work has explored the use of time-frequency-based speech representations such as spectrograms and log-mel spectrograms as input for deep learning architectures to classify depression from audio. Srimadhur et al. [28] investigated spectrograms as well as raw waveform as input to CNN-based network on a subset of DAIC-WOZ dataset in speaker-dependent fashion. Moreover, in the study by Ma et al. [29], a

CNN-LSTM based architecture was explored that extracted discriminative features from mel-spectrograms using 1d convolution in the first layer. A random sampling strategy was also proposed to mitigate the data imbalance issue associated with the DAIC-WOZ dataset. The majority voting of the labels for segments of speech coming from an individual is used for depression prediction for an individual. In a recent study by Vazquez-Romero et al. [30], an ensemble of 1d-CNN networks is used with mel-spectrograms as input features. The label for an individual is generated by the mean of the segment level probabilities for each constituent network in the ensemble, and the ensemble labels are averaged to yield a final label for the individual. This ensemble technique demonstrated appreciable improvements in recognition performance over hand-crafted features based on SVM classification and other single deep learning-based networks.

Multiple instance learning (MIL) is the apt choice when a single label is available for a group of utterances as in Depression classification problem [31]. The majority of the approaches in literature exploiting MIL architecture works by generating labels for individual segments and averaging them to yield a final label for the whole utterance. This is done using a network that shares parameters with all the segments of an utterance [32, 33]. However, the inherent problem with the MIL framework for depression classification is that not all the segments of the utterance exhibit depression-related characteristics, with the majority of the segments being in a neutral emotional state. As such, false labels are predicted quite often due to the majority of neutral state segments.

Motivated by these limitation of the extant modelling methodologies, we developed a Tensor-based approach to extract shared and discriminative features from multiple segments of an utterance. Tensor factorizations provide a natural method for analyzing common information spread across modes of a tensor [34]. Utilizing this aspect, we use tensor factorization in conjunction with neural network-based learning to address the multiple-instance learning in a novel framework. Furthermore, the utterance level tensor core generated by the feature extraction block is passed on to an attention mechanism to generate the utterance level attentive feature. Statistic pooling of attentive representations is performed to extract bag-level features, which are classified using fully connected layers. This mitigates the dependence on average/max pooling output labels for individual segments for utterance level prediction, thus countering the inherent issue of traditional MIL frameworks. The proposed tensor based MIL approach for depression classification outperforms several state-of-the-art methodologies and provides a promising avenue for robust depression classification from speech signals.

## 2 Materials and methods

### 2.1 Tensor preliminaries

We review the introductory multilinear algebra, which is necessary to understand Tucker decomposition. A detailed, comprehensive review of tensor algebra can be found in [34, 35]. Sticking with the notations used in tensor literature, a vector is denoted by a lowercase letter (e.g. 'a'), a matrix with an uppercase letter (e.g. 'A') and tensors of order three or more by calligraphic letters(e.g. '$\mathcal{A}$').

Tensors are multidimensional arrays e.g. $\mathcal{X} \in \mathbb{R}^{I_1 \times I_2 \times \cdots \times I_n}$, where $n$ is the number of modes in the tensor, also referred to as order of the tensor, which may correspond to space, time, frequency, trials, utterances etc and $I_n$ specifies the dimensionality of the mode corresponding to $n$th mode of the tensor $\mathcal{X}$. Tensor manipulation often requires its reshaping to matrix form, and one such particular reshaping is called mode-n matricization or unfolding. For a third order tensor $\mathcal{X} \in \mathbb{R}^{I_1 \times I_2 \times I_3}$, mode-n matricization is achieved by fixing one index and varying the other two. It is denoted by $\mathbf{X}_{(n)} \in \mathbb{R}^{I_n \times (I_1 \times I_2 \times \cdots \times I_{n-1} \times I_{n+1} \times \cdots \times I_N)}$, where the column vectors of

$\mathbf{X}_{(n)}$ are the mode-n vectors of $\mathcal{X}$. For $N$ matrices, one corresponding to each mode, we denote it using a superscript in parenthesis, example $\mathbf{U}^{(n)}$.

Mode-n multiplication of a tensor $\mathcal{X}$ with a matrix $\mathbf{U}$ is obtained by multiplying all the vector fibers of a mode-n matrix with the matrix $\mathbf{U}$. It is denoted as $\mathcal{Y} = \mathcal{X} \times_n \mathbf{U}$, and in matrix form it can be written as

$$\mathcal{Y}_{(n)} = \mathcal{X}_{(n)}.\mathbf{U} \tag{1}$$

Multilinear subspace requires the understanding of multilinear projections as a tensor subspace is defined as a mapping from high-dimensional space to a low-dimensional space [36]. Considering the general case of higher order tensors, an Nth order tensor $\mathcal{X} \in \mathbb{R}^{I_1 \times I_2 \times \cdots \times I_N}$ resides in the tensor space $\mathbb{R}_1 \otimes \mathbb{R}_2 \otimes \cdots \mathbb{R}_N$, where $\mathbb{R}_1, \mathbb{R}_2, \cdots \mathbb{R}_N$ denotes real vector spaces and $\otimes$ represents the tensor outer product (for details see [34]). As such, the tensor space for N order tensors consists of the outer product of $N$ vector spaces $\mathbb{R}_n$, $n \in 1, 2, \cdots, N$. A tensor $\mathcal{X} \in \mathbb{R}^{I_1 \times I_2 \times \cdots \times I_N}$ can be projected onto a lower dimensional tensor $\mathcal{Y} \in \mathbb{R}^{P_1 \times P_2 \times \cdots \times P_N}$, where $P_n \leq I_n$ using N projection matrices $\mathbf{U}^{(n)} \in \mathbb{R}^{I_n \times P_n}$, one corresponding to each mode of the tensor.

$$\mathcal{Y} = \mathcal{X} \times_1 \mathbf{U}^{(1)^T} \times_2 \mathbf{U}^{(2)^T} \cdots \times_N \mathbf{U}^{(N)^T} \tag{2}$$

**2.1.1 Tucker decomposition.** Tucker decomposition of a third order tensor $\mathcal{Y} \in \mathbb{R}^{I_1 \times I_2 \times I_3}$ is defined as a multilinear transformation of a core tensor, generally of small size and dense, by the factor matrices corresponding to each mode of the tensor [34, 37].

$$\mathcal{Y} = \mathcal{X} \times_1 \mathbf{U}^{(1)} \times_2 \mathbf{U}^{(2)} \times_3 \mathbf{U}^{(3)} \tag{3}$$

Here, $\mathbf{U}^{(1)} \in \mathbf{R}^{I_1 \times P_1}$, $\mathbf{U}^{(2)} \in \mathbf{R}^{I_2 \times P_2}$ and $\mathbf{U}^{(3)} \in \mathbf{R}^{I_3 \times P_3}$ corresponds to the subspaces along mode-1, mode-2 and mode-3 respectively The subspaces consists of the basis vectors obtained from matrix unfolding along each mode of the tensor. Tucker decomposition has the constraint of orthogonality and ordering on the core tensor and factor matrices, while other constraints such as non-negativity, sparsity, etc. can also be imposed.

A matrix representation of the tucker decomposition, in general case, can be achieved by matricizing $\mathcal{Y}$ and $\mathcal{X}$ as [38]

$$\mathbf{Y}_{(n)} = \mathbf{U}_{(n)}.\mathbf{X}_{(n)}(\mathbf{U}^{(n+1)} \otimes \cdots \otimes \mathbf{U}^N \otimes \mathbf{U}^{(1)} \otimes \cdots \otimes \mathbf{U}^{(n-1)}) \tag{4}$$

where $\otimes$ denotes the Kronecker product. The decomposition can also be written as a linear combination of $\prod_{n=1}^{N} I_n$ rank one tensors.

$$\mathbf{Y} = \sum_{i_1=1}^{I_1} \sum_{i_2=1}^{I_2} \cdots \sum_{i_N=1}^{I_N} \mathcal{X}(i_1, i_2, \cdots, i_N) \mathbf{u}_{i_1}^{(1)} \circ \mathbf{u}_{i_2}^{(2)} \circ \cdots \circ \mathbf{u}_{i_N}^{(N)} \tag{5}$$

## 2.2 Dataset and preprocessing

For the task of depression classification from speech signals, we use the audio modality from the Distress Analysis Interview Corpus-Wizard of Oz (DAIC-WOZ), which is a subset of the larger corpus DAIC [39] and was introduced in the Audio/Visual Emotion Challenge (AVEC) 2016 [40]. The dataset consists of clinical interviews conducted between a participant and a virtual interviewer *ellie* which was controlled by a human interviewer remotely. The dataset was collected with the motive to augment the diagnoses of psychological conditions such as

**Table 1. Distribution of male and female participants across train, validation and test partitions of the DAIC--WOZ depression dataset.**

| Partition | Male | Female |
|---|---|---|
| Train | 63 | 44 |
| Validation | 16 | 19 |
| Test | 23 | 24 |

stress, anxiety, depression, etc., through automatic computer applications based on verbal and non-verbal indicators. It consists of audio, facial geometry features as well as text transcriptions of the interviews. Table 1 shows the distribution of participants according to gender for the train, validation and test partitions. The dataset is recorded in English from a population of 189 subjects comprising 146 depressed subjects and 43 healthy controls. The duration of the audio ranges from 7-33 min (average 16 minutes). Each participant's audio file has been given a PHQ-8 score by the psychologist, which denotes the severity of depression, with 0 being no depression to 22 being severely depressed. Also, a binary PHQ-8 score is also provided, which classifies participants as depressed/not-depressed. Furthermore, the train-development-test split provided by the AVEC 2016 challenge divides the dataset into partitions comprising of 118, 24, 47 participants in the train, development, and test set, respectively.

Since the virtual interviewer's speech is not a part of the analysis, a silence region-based segmentation technique from the Python library *PyAudioAnalysis* [41] is employed to segment out the participant's speech and discard the speech segments from the virtual interviewer as it doesn't contain any emotion information. Also, the speech segments produced are of different duration, and deep learning techniques such as CNN and TFNN [42] require equal length input, so the speech segments are either zero-padded or truncated to 7 secs duration. The sampling rate of the speech signal is 16 kHz.

## 2.3 Methodology

This section discusses the Tensor Factorization-based Multiple-Instance Learning Technique, which is used for the classification of depression versus normal speech from multiple utterances of a single speaker. Furthermore, an utterance level attention followed by a statistics pooling layer [43] is employed to extract temporal features in the subsequent layers of the network. Moreover, a standard Multiple-Instance Learning (MIL) network based on Convolution layers is also discussed, which serves as a baseline for comparing results.

**2.3.1 CNN and 2D TFNN based MIL framework.** Multiple Instance Learning with CNN as a base architecture has been explored in many previous works [44, 45]. As such, we have used this architecture as a baseline in our work. The base CNN architecture comprises of 3 feature learning blocks followed by vectorization of the deep features and classification using a sigmoid layer. Each feature learning block comprises a 2D convolution layer, a batch normalization layer, an activation layer, and a max-pooling layer. The convolution layer extracts local features with the help of trainable kernels. Batch normalization forces the mean of the features over the entire batch to be centered at zero with unit variance. The normalized features are passed through an activation function (ELU in our work). Finally, a max-pooling layer is employed to reduce the size of the feature maps obtained, keeping the relevant information only. Given a bag of utterances belonging to a speaker, the base CNN architecture is employed on each of the utterances to yield a label for each utterance. A global max-pooling of the labels yields the final label for the bag of utterances.

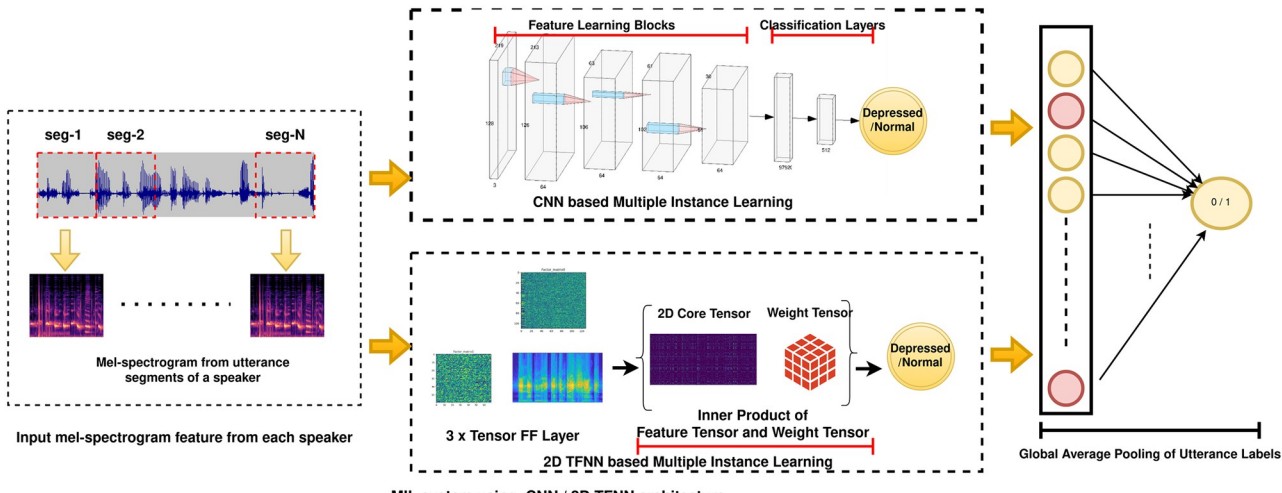

**Fig 1. MIL technique using CNN (top) and TFNN (bottom) as base architectures.** The input to the architectures is 2D Mel-spectrogram tensor, generated from speech utterances of a speaker. For the CNN architecture, stacks of three feature learning blocks followed by vectorization and dense layers is depicted. For TFNN, stacks of three 2-D Tensor FF layer followed by a Tensor Sigmoid layer is depicted. Finally, global average pooling of utterance labels is shown.

A 2D TFNN architecture [46] is employed as a base network for the MIL, similar to the CNN architecture. The 2D TFNN base receives mel spectrograms extracted from speech utterances as input. The factor matrices corresponding to the time and frequency modes extract the core feature tensor from the input tensors. Four consecutive Tensor FF layers yield the final feature tensor, which is then used to generate a class probability by doing an inner product with a weight matrix of the exact dimensions as the feature tensor. Fig 1 shows the end-to-end system for 2D CNN and 2D TFNN based MIL architecture.

**2.3.2 3D TFNN architecture as feature extractor for MIL.** The 3D TFNN architecture was introduced in [46] for emotion recognition from speech. The 3D TFNN serves as a natural framework for Multiple Instance Learning as the core idea of Tensor Factorization is capturing the shared information across different modes of a tensor. As such, given a bag of utterances belonging to a speaker, the utterances are first converted to the 2D speech representations such as mel-spectrograms of dimensions $I_{freq} \times I_{time}$. The mel-spectrograms for each utterance are stacked along the 3rd dimension to form a 3D-tensor of dimensions $I_{freq} \times I_{time} \times I_{utter}$ representing the bag of utterances. The 3D tensor is passed through successive Tensor Factorization layers to obtain the deep feature tensors. Finally, a tensor sigmoid layer, comprising a weight tensor of the same size as the deep feature tensor, is utilized to get the probability for the bag of utterances.

The 3D TFNN architecture for Multiple Instance Learning benefits from not repeating the same architecture individually on each utterance as in conventional CNN-based MIL systems. Moreover, the probability generated by the 3D TFNN represents the entire bag as opposed to CNN-based MIL, where a global max-pooling of the labels generates a bag-level label. This comes from the inherent capability of Tensor Factorization-based feature extraction. The shared information across mel-spectrograms of utterances for an individual is utilized to conclude the label for that particular speaker. In contrast, the utterance level information is independent in conventional MIL systems, and no shared information across utterances is utilized. Fig 2 shows the proposed end-to-end Tensor factorization based approach for MIL.

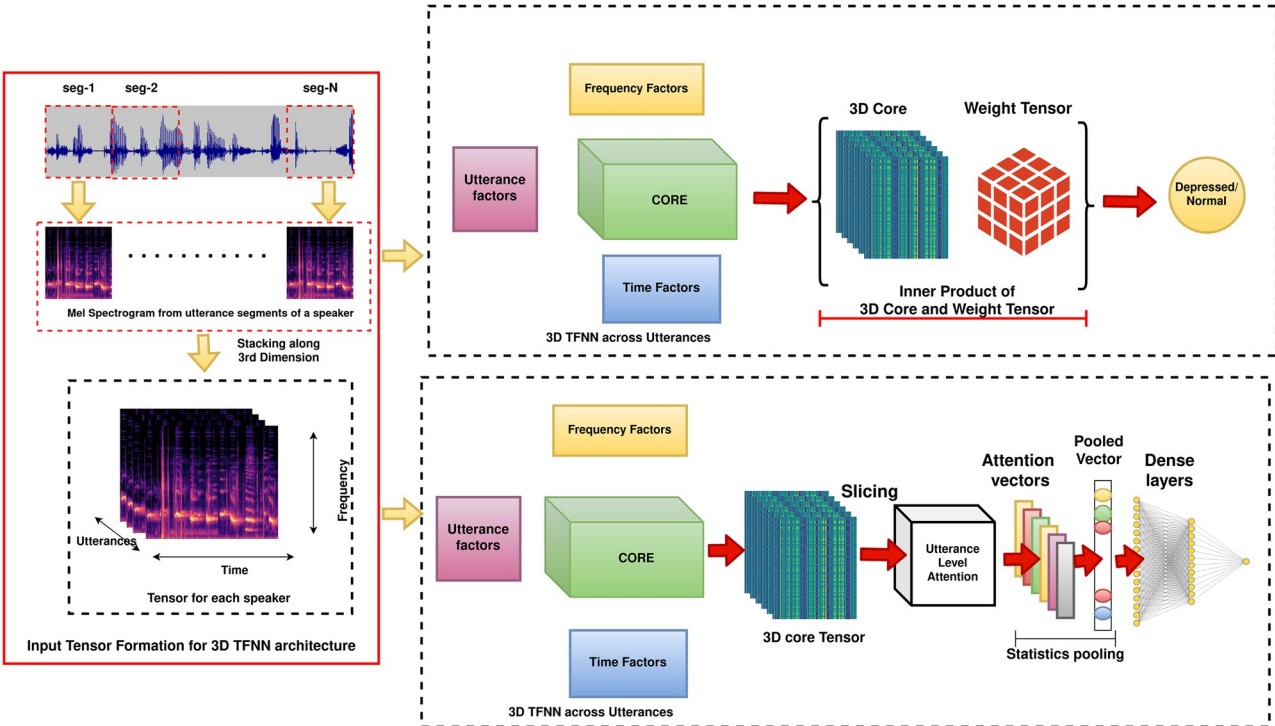

**Fig 2. MIL technique using 3D TFNN and 3D TFNN + Utterance level attention as base architectures.** The input to the architectures is Mel-spectrogram tensor, generated by stacking mel-spectrograms of utterances by a speaker along the third dimension. For the 3D TFNN, stacks of 3D Tensor FF Layer followed by a Tensor Sigmoid Layer is depicted. For the 3D TFNN + Utterance level attention, the stacks of 3D Tensor FF Layer is followed by an utterance level attention layer. A statistics pooling layer aggregates information from the attentive feature vectors of utterances followed by a dense layer for classification.

**2.3.3 3D TFNN with utterance level attention.** In this technique, the 3D TFNN described in 2.3.2 is utilized to extract deep tensor features from 3D tensor representations of bags of utterances. The feature tensor now comprises utterance level representations stacked along the third dimension of the feature core tensor. For each 2D slice of the 3D feature tensor, an utterance level attentive feature representation is generated using the following attention mechanism.

*2.3.3.1 Attention layer.* The attention layer used in our work is based on the attention proposed in [47]. The attention layer takes in a sequence of high-level feature vectors, focuses on the depression-related parts employing attention weights, and generates an utterance level attention feature vector representing the depression-related frames of the input sequence. Given a 2D slice $\mathbf{H} \in \mathbf{R}^{I_2 \times I_3}$ of 3D feature tensor tensor $\mathcal{X} \in \mathbf{R}^{I_1 \times I_2 \times I_3}$, where $I_1, I_2, I_3$ represents the number of utterances, number of mel filter bands and number of frames respectively, normalized attention weights are first computed using a softmax function as described in equation -

$$\alpha_t = \frac{\exp(W.h_t)}{\sum_{t=1}^{T} \exp(W.h_t)} \tag{6}$$

where $t \in (1, 2, \cdots, T)$, $T$ being the total number of frames in the feature tensor slice and $h_t$ being a feature vector belonging to the $t$th frame. The utterance level feature vector is obtained

by taking the weighted sum of the attention weights with $h_t$ as following -

$$\mathbf{c} = \sum_{t=1}^{T} \alpha_t h_t \qquad (7)$$

*2.3.3.2 Statistics pooling.* The statistics pooling was first introduced in [43] for extracting utterance level statistics from frame-level features embeddings generated using a Time Delay Neural Network for speaker verification tasks. In our proposed architecture, statistics pooling is employed to extract bag level statistics—mean and standard deviation from the utterance level attentive feature vectors. As such, the output of the statistics pooling layer aggregates the relevant discriminative information obtained from several speaker utterances and provides a unified feature for further classification objectives. Given a set of attention feature vectors $C = (c_1, c_2, \cdots, c_{I_1})$ and $c \in \mathbf{R}^{I_2}$, obtained as described in section 2.3.3.1, where $I_1$ represents the number of utterances in the bag, the statistics pooling is calculated using *mean*, which is the average and *var*, which is the variance -

$$\mu = mean(C) \qquad (8)$$

$$\sigma = var(C) \qquad (9)$$

This results in a pooled feature vector of dimensions $\mathbf{R}^{2 \times I_2}$, with $\mu$ and $\sigma$ concatenated for each entry of $c$.

*2.3.3.3 Fully connected layer.* The output from the statistics pooling layer contains the aggregation of information across several utterances of a speaker. The pooled feature vector is passed to a fully connected network, having two layers to reduce the dimensionality and extract additional high-level features. Finally, the output of the fully connected layers is passed on to the last layer with sigmoid activation to generate the classification probability of being depressed/ normal.

**2.3.4 Experimental setting.** The four architectures—baseline CNN-MIL, TFNN-MIL, 3D TFNN, and 3D TFNN+Attention, are evaluated on the DAIC-WOZ dataset for Depression classification. For tensor formation, a set of utterances or bag sizes in the range [10, 60] are selected from each speaker. Thus multiple tensors are formed for each speaker considering multiple bags formed because of the bag size chosen without repetition of utterances. For the training scenario, each individual bag of utterances is considered coming from a new speaker bearing the same label as all the other children bags of the parent speaker, thereby generating a large number of tensors for training. However, for the testing scenario, the label for the parent speaker is calculated by averaging the predicted probability of all the children bags and comparing the final averaged probability against a threshold. The threshold is calculated from the ROC curve generated using the validation data.

Mel spectrograms are computed from the speech segments to be used as input for the Tensor Factorized Neural Network and baseline CNN architecture. For the computation of mel spectrograms, the speech segments are first windowed using a hamming window of 2048 samples with a shift of 512 samples. The windowed signal is used to compute Short-Time Fourier Transform (STFT). The magnitude spectrogram obtained from STFT is then passed through a mel-scale to obtain the filterbank energies. A log operator is finally used to get the log-mel spectrogram.

For baseline CNN architecture, the number of filters in the first and second feature learning block is 64 with a kernel size of $3 \times 3$ and a shift of 1. For the third feature learning block, the number of filters is 128 with kernel size $2 \times 2$. The activation function used in all feature

learning blocks is *ELU* and a max-pooling with kernel size of $2 \times 2$ is used. The feature maps generated after the third feature learning block is vectorized and passed through a fully connected network with sigmoid non-linearity in its last layer to generate probabilities for the depressed versus non-depressed categories.

For the TFNN-MIL system, the base architecture consists of four consecutive 2D Tensor Feed Forward layers. The features dimension produced from the Tensor FF layers are respectively $120 \times 210$, $110 \times 200$, $100 \times 180$ and $80 \times 160$. The output from the fourth Tensor FF layer is used to calculate logits using an inner product with a weight tensor of dimensions $80 \times 160$. Finally, the logits are passed through the activation function to yield utterance segment-level probabilities. This base architecture is repeated for all the instances in the bag, and a final global average pooling of the probabilities generates the bag level probability.

For 3D TFNN architecture, the input tensor is of size $num_{utter} \times 128 \times 219$ where the dimensions refer to the number of utterances, mel filters, and the number of time frames, respectively. The input mel-spectrogram tensor is passed through two 3D tensor feed-forward layers where the core tensors are of size $num_{utter} \times 120 \times 200$ and $num_{utter} \times 100 \times 180$ respectively. The activation function used in both the Tensor FF layers is *RELU*. The feature tensor obtained after the second Tensor FF layer is fed to a Tensor sigmoid layer. The output of the inner product of the feature tensor with a trainable weight tensor of the same size is passed through a sigmoid non-linearity to generate class probability.

In the case of 3D TFNN+ Attention architecture, two 3D tensor FF layers, as used in 3D TFNN architecture above, extract discriminative feature tensor of the size $num_{utter} \times 100 \times 180$. The utterance level attention mechanism generates utterance level feature vectors of dimensions $num_{utter} \times 100$. This feature sequence is passed to a statistics pooling layer generating a feature vector of dimensions $\mathbf{R}^{200}$, which is passed through two fully connected layers of dimensions 256, 256 and a last layer having sigmoid non-linearity to generate class probability for the bag of utterances.

## 3 Results

The four architectures—baseline CNN-MIL, TFNN-MIL, 3D TFNN, and 3D TFNN+Attention, are trained and evaluated on the DAIC-WOZ dataset using the following metrics— weighted accuracy, unweighted accuracy, and F1-score. Since the dataset is highly imbalanced, unweighted accuracy and F1-score becomes the apt choice to highlight the true prediction capability of the models. Moreover, another inherent issue with class imbalanced datasets is threshold-moving, which makes the default threshold of 0.5 for binary classification problems shift. For our work, we have utilized the optimal threshold calculated from the ROC curve on the validation dataset, which is the development partition of the dataset. The optimal threshold is then used to generate labels for the probabilities predicted for the test set.

As seen from the Table 2, the 3D TFNN and 3D TFNN + Attention architecture outperforms the baseline CNN-MIL system by a considerable margin of 16.67% and 17.2%

**Table 2. Recognition accuracies in terms of Weighted Accuracy (WA) and Unweighted Accuracy (UA) and F1-scores for different tensor based techniques for test set of Daic-Woz dataset.**

| Method | Single Utterances | | Speaker Level | | |
|---|---|---|---|---|---|
| | WA(%) | UA(%) | WA(%) | UA (%) | F1-score (Normal, Depressed) |
| CNN MIL | 54.40 | 55.65 | 51.06 | 54.87 | 0.56,0.43 |
| TFNN MIL | 60.00 | 62.52 | 65.95 | 71.64 | 0.70,0.60 |
| 3D TFNN | 59.20 | 65.17 | 74.47 | 71.54 | **0.81,0.60** |
| 3D TFNN + Att | 60.40 | 61.06 | 72.34 | **72.07** | 0.78,0.60 |

respectively in terms of UA. This justifies that Tensor Factorized Neural Networks are more suitable for MIL-based systems due to their common information capturing capability amongst several modes of the tensor input. Moreover, the 3D TFNN+Attention system provides a balance of overall accuracy to average of class accuracies. This becomes important for imbalanced datasets where the model's chances of fitting towards the majority class are always high. Moreover, in terms of F1-score, 3D TFNN outperforms other techniques and reaches the state-of-the-art.

Fig 3 presents the confusion matrices for the four architectures on the test set of the DAIC-WOZ dataset, taking 30 utterances per tensor. It is evident from the confusion matrix in Fig 3d that 3D TFNN+Attention architecture can balance the model toward both depressed and non-depressed categories, followed by 3D TFNN architecture. This supports our proposal of using utterance level attention to generate attentive feature vectors per utterance segment. Moreover, the impact of the number of utterances per tensor on the recognition performance of the model is assessed in Fig 4. The range for the number of utterances per tensor is considered in the interval [10, 60]. The figure is plotted using b-spline interpolation [48] to account for the fewer data points and getting a smooth curve. As is evident from the graph, the model performs best when 30 utterances are chosen per tensor. Also, the performance shows a gradual decline in the accuracy when the number of utterances per tensor is increased. This may be because redundant information apart from the desired objective is also being captured with increasing utterances, which accounts for increased confusion and decreased accuracy.

## 3.1 Comparison with State-of-the-Art

Several studies have utilized Daic-WoZ Depression dataset for unimodal as well as multimodal depression recognition [20, 49]. Since in this investigation, we have considered only the audio modality, the performance is compared with other studies using audio modality only. Moreover, few studies have reported the final results which are limited on the development partition of the dataset. More importantly, our work utilizes the test set as the unseen data; we compare with similar works reporting results on test partition. Also, the published studies are

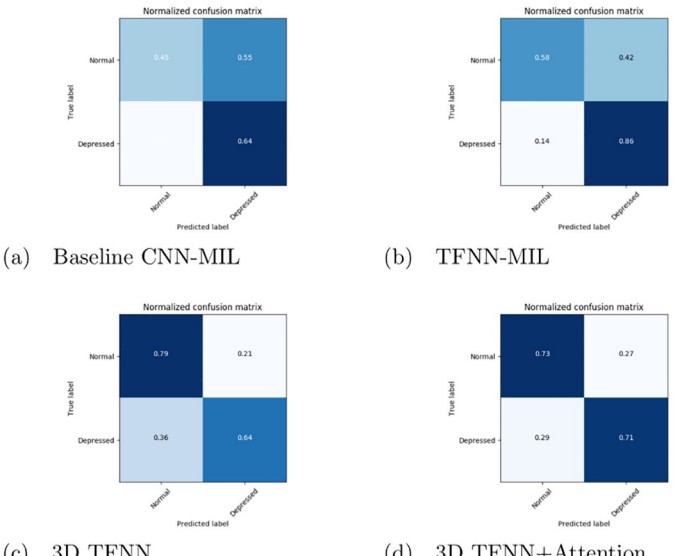

(a)    Baseline CNN-MIL

(b)    TFNN-MIL

(c)    3D TFNN

(d)    3D TFNN+Attention

**Fig 3. Normalized confusion matrix for the test set of DAIC-WOZ depression dataset for the three architectures —Baseline CNN-MIL, TFNN-MIL, 3D TFNN and 3D TFNN+Attention.**

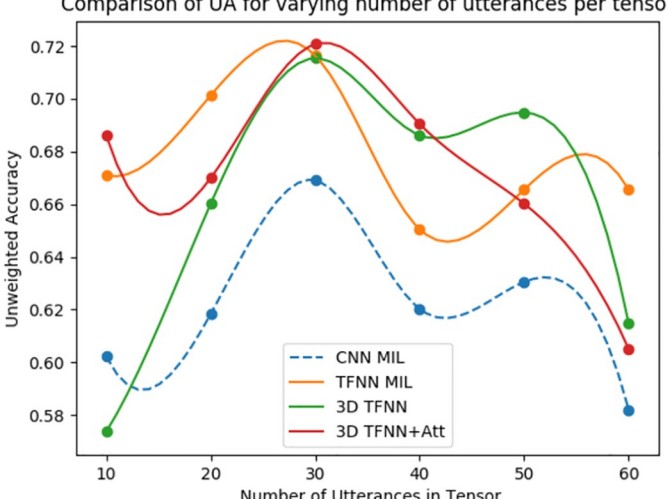

**Fig 4. Comparison of unweighted accuracy for varying number of utterances per tensor for the architectures CNN-MIL, TFNN-MIL, 3D TFNN and 3D TFNN+Attention.**

segregated upon the metrics used to give a fair comparison and restricted to the ones which have used accuracy and F1-score as metrics have been included for comparison.

Table 3 presents the state-of-the-art techniques for Depression recognition from speech utterances using the DAIC-WOZ dataset. Valstar et al. [40] provided the baseline results for the DAIC-WOZ dataset using both the audio and video modality. Our novel implementation outperforms the baseline by 0.21 for the mean F1-score for the audio modality scenario. Previously, Ma et al. [29] utilized a combination of CNN and LSTM networks to extract high-level features from raw speech representations and uses a random sampling strategy to balance out the examples between depressed and normal classes. In contrast, our investigation uses a weighted loss function to alleviate the imbalance of classes and thereby incorporate all the training speakers during model training. As such, our proposed architecture achieves an overall performance gain of around 9% in terms of accuracy.

## 3.2 Discussion

Several features have been investigated in literature for depression diagnosis from speech utterances. This study focused on mel-spectrograms for two reasons. First, mel-spectrogram has proven to contain para-linguistic information present in speech utterances such as emotional states [50], cough [51] etc. Secondly, spectrograms provide a natural 2D tensor form for speech utterances. The proposed Tensor-Based MIL techniques tries to exploit the time-

**Table 3. Comparison with the state-of-the-art techniques on the test partition of DAIC-WOZ dataset in terms of Weighted Accuracy(WA), Unweighted Accuracy (UA) and F1-scores.**

| sl.no | Method | Year of Publication | Accuracy | | F1 score | | |
|---|---|---|---|---|---|---|---|
| | | | WA | UA | Depressed | Normal | Mean |
| 1. | Valstar et al. (AVEC base) | 2016 | - | - | 0.41 | 0.58 | 0.495 |
| 2. | Ma et al. (DepAudioNet) | 2016 | 0.65 | - | 0.52 | 0.70 | 0.610 |
| 3. | Romero et al. (Ensemble) | 2020 | 0.72 | - | 0.63 | 0.78 | 0.705 |
| 4 | **3D TFNN (proposed)** | - | **0.745** | **0.715** | **0.60** | **0.81** | **0.705** |

frequency information spread across several utterances of a speaker. The 3D TFNN extracts shared information across the mel-spectrograms of a speaker, thus trying to model the temporal information spread across multiple utterances in an interview setting. The 3D core tensor, which is the feature tensor, is comprised of the coefficients of interactions across the subspaces corresponding to each of the modes- time subspace, frequency subspace and utterance subspace. Moreover, when using utterance-level attention, the model tries to extract more relevant information pertaining to depression from each utterance by the means of self-attention. This in turn refines the feature extraction process by producing attentive feature vectors for each utterance in the tensor. To aggregate the information extracted using attention layers, statistics pooling is used, which generates a combined feature vector for all the utterances in the tensor. The proposed techniques are computationally efficient as using Tensor Factorization based architecture significantly lowers the number of trainable parameters [46].

## 4 Conclusion

In this work, we present a tensor-based architecture for the task of Multiple Instance Learning when a collection of utterances for a speaker is available, and inferences about the speaker label have to be drawn using the feature set from utterances. The conventional MIL architectures such as the baseline CNN-MIL system described in Fig 1 suffer from the inherent drawbacks of not considering relative and shared information across the utterances in a bag. These techniques rely on inferring labels for individual utterances and finally averaging or max-pooling the labels to infer the speaker-level labels. The tensor-based architectures solve this problem by considering the utterances as the third mode in addition to the time and frequency modes in speech spectrograms. As such, TFNNs, by its rich mathematical framework, try to capture the shared information across the utterances of a bag by tensor factorization where the input tensor is projected over three subspaces—time subspace, frequency subspace, and utterance subspace. This helps to leverage the shared information and generate a single speaker/bag level probability for the specified task. To this end, we have implemented two tensor MIL architectures—3D TFNN and 3D TFNN+Attention. Comparison with the state-of-the-art proves that both these novel techniques are effective in capturing depression-related information across bags of utterances. Moreover, additional analysis on the optimal number of utterances per bag is also presented to shed light on the model performance when using varying bag sizes.

## Author Contributions

**Conceptualization:** Sandeep Kumar Pandey, Hanumant Singh Shekhawat, S. R. M. Prasanna, Shalendar Bhasin, Ravi Jasuja.

**Investigation:** Sandeep Kumar Pandey.

**Methodology:** Sandeep Kumar Pandey.

**Project administration:** Hanumant Singh Shekhawat.

**Resources:** Hanumant Singh Shekhawat.

**Software:** Sandeep Kumar Pandey.

**Supervision:** S. R. M. Prasanna, Shalendar Bhasin, Ravi Jasuja.

**Validation:** Sandeep Kumar Pandey, Ravi Jasuja.

**Visualization:** Sandeep Kumar Pandey.

**Writing – original draft:** Sandeep Kumar Pandey, Ravi Jasuja.

**Writing – review & editing:** Hanumant Singh Shekhawat, S. R. M. Prasanna, Shalendar Bhasin, Ravi Jasuja.

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
