## [Decision Letter · Decision Letter 0]

31 Mar 2022

PONE-D-22-04732A deep tensor-based approach for automatic depression recognition from speech utterancesPLOS ONE

Dear Dr. Pandey,

Thank you for submitting your manuscript to PLOS ONE. After careful consideration, we feel that it has merit but does not fully meet PLOS ONE’s publication criteria as it currently stands. Therefore, we invite you to submit a revised version of the manuscript that addresses the points raised during the review process.

We look forward to receiving your revised manuscript.

Kind regards,

Dhananjay Singh, Ph.D.

Academic Editor

PLOS ONE

Journal Requirements:

2. Acknowledgments Section: Move New Information to the Financial Disclosure:

"Thank you for stating the following in the Acknowledgments Section of your manuscript: 

[This research was supported under the India-Korea joint program cooperation of science

and technology by the National Research Foundation (NRF) Korea

(2020K1A3A1A68093469), the Ministry of Science and ICT (MSIT) Korea and by the

Department of Biotechnology (India) (DBT/IC-12031(22)-ICD-DBT).]

 [This research was supported under the India-Korea joint program cooperation of science and technology by the National Research Foundation (NRF) Korea (2020K1A3A1A68093469), the Ministry of Science and ICT (MSIT) Korea and by the Department of Biotechnology (India) (DBT/IC-12031(22)-ICD-DBT).]

3. Please ensure that you refer to Figure 1 in your text as, if accepted, production will need this reference to link the reader to the figure.

Additional Editor Comments:

-In the related work, need to include more recently published work with explanation of what is the role of the deep learning research in depression analysis?

-I suggest to more detailed explanation for Evaluation based on framework for automated depression classification and the compression with others recent work.

- Please explain in briefly and move their explanation in a paragraph explaining what each table and figures refe to.

- It will be helpful to have a few lines explaining what is inside the features chosen and why a particular fusion method is suitable for the particular depression analysis.

Reviewers' comments:

Reviewer's Responses to Questions

**Comments to the Author**

1. Is the manuscript technically sound, and do the data support the conclusions?

Reviewer #1: Yes

Reviewer #2: Yes

2. Has the statistical analysis been performed appropriately and rigorously? 

Reviewer #1: Yes

Reviewer #2: Yes

3. Have the authors made all data underlying the findings in their manuscript fully available?

Reviewer #1: No

Reviewer #2: Yes

4. Is the manuscript presented in an intelligible fashion and written in standard English?

Reviewer #1: Yes

Reviewer #2: Yes

5. Review Comments to the Author

Reviewer #1: The paper is strong and innovative. However, these areas should be improved:

1- Please discuss the work of U of Michigan team on PRIORI which uses pitch to predict mood of patients with bipolar disorder. Mccinis, Khorram, and others are among the authors. Please compare your work with theirs and let us know how your work has enhanced their work.

2- Please add the n to all your tables.

3- Quality of figures are not acceptable. These figures are hard to follow.

4- Too many abbreviations. Please reduce the use of abbreviations to those very nessasary.

5- PHQ8 and PHQ 9 differ, and their results should not be combined. Why not limiting the results to 8 items.

6- If PHQ is used, why do you discuss and cite Hamilton measure too?

7- We need tables that descripbe the participants.

8- Show us correlation between all variables. (Pearson r)

9- Please use MDD for major depression. DOes not need the letter "S".

Reviewer #2: The author's article on “ A deep tensor-based approach for Automatic Depression Recognition(ADR) from Speech utterances” is interesting for looking into the dimension of human mental health which is currently an important topic in every age groups of mankind and its analysis through speech processing presently societal need. The authors are appreciated for their intuitive knowledge in speech analysis and current technology exploration.

Article is acceptable for minor corrections

Minor Suggestions :

The author's article on “ A deep tensor-based approach for Automatic Depression Recognition(ADR) from Speech utterances” is interesting for looking into the dimension of human mental health which is currently an important topic in every age groups of mankind and its analysis through speech processing presently societal need. The authors are appreciated for their intuitive knowledge in speech analysis and current technology exploration.

With minor correction the article is very much acceptable for the journal.

1. The title of the article says depression recognition so the authors are only specific parameters based patterns generated to Recognition or authors may look the contextually fit then instead recognition detection is more suitable.? Justify with your answer.

2. The Word GIVE must be justified or reference adds for abbreviation if taken from research sources. Give the significant reasons for capitalization of the word

3. In page 1 line no. 6 of Introduction section Major Depression Disorder(MSDD) and Multi system depressive disorder must be given proper relation and both abbreviations cannot be interchangeable hence look into these point and justify your comments.

4. In Line No.10 page 2 the numbers added with the symbols of percentage and again word used so redundancy in information may be corrected. Ex: 80% percentage is not correct 80 percent or 80% along correct. Kindly justify your action on this modification

5. Line 18 of page 2 “P.H.Q” abbreviated word usability in the entire document must be consistent.

6. Differentiation of the depression detection and depression recognition must be distinguished while using the work in the article. The usability of the word must be maintained in a consistent manner.

7. Appreciate the authors with more grammatically correct and with simple sentence breaks made throughout the article in a consistent manner.

8. Authors can justify properly the use of Abbreviations separated full stops and some places without them. Entire articles use a uniform process.

9. Page No. 8 Section 3.3.2 given equations need to be numbered.

10. Page no. 9 and content in Table 1. Captions of the Row must be added to the percentage symbols as the accuracy is measured with ratios.

11. Authors suggested rewriting the Abstract and Conclusions with simple sentences by appropriate breaks and conveying the authors view properly.

12. The technical concepts are good and explanation flow is appreciable.

With this minor correction the article is acceptable.

6. PLOS authors have the option to publish the peer review history of their article (what does this mean?). If published, this will include your full peer review and any attached files.

Reviewer #1: **Yes: **Shervin Assari

Reviewer #2: No

---

## [Author Response · Author response to Decision Letter 0]

25 May 2022

1

I. COMMENTS FROM REVIEWER 1

A. Comment

Please discuss the work of U of Michigan team on PRIORI which uses pitch to predict mood

of patients with bipolar disorder. Mccinis, Khorram, and others are among the authors. Please

compare your work with theirs and let us know how your work has enhanced their work.

Reply: We like to thank the reviewer for the suggestion. We have cited the paper and have added

discussion in the introduction section. The work in ”The PRIORI Emotion Dataset: Linking Mood

to Emotion Detected In-the-Wild” has investigated emotion in speech as an intermediary feature

for monitoring Bipolar Disorder over depressed and manic states of an individual. The authors

have proposed a new dataset collected from smartphone speech recordings and assessed several

parameters such as PCC, CCC, RMSE etc to prove the robustnes of the feature and deep learning

methods in identifying emotional activation and valence.

Compared to this suggested work, we have explored depression recognition in an individual

based on his speech utterances. The architecture proposed tries to capture shared information

across several utterances of a speaker spread temporally across the interview conversation. The

emotion aspect as suggested by the reviewer is very interesting and is part of our future work

in extending this study.

B. Comment

Please add the n to all your tables.

Reply: Thank you for the suggestion. However, we are not clear regarding the expectation of the

reviewer in this comment. We have tried to stick to PLOS ONE Journal formatting guidelines

to the best.

C. Comment

Quality of figures are not acceptable. These figures are hard to follow.

Reply: We deeply regret the inconvenience in understanding the figures in the current form.

To incorporate the reviewer’s suggestion, we have added a brief description of the components

of the figure in the caption to understand the information flow along various components. We

have updated the figures to represent the end-to-end process starting from speech signal to label

generation.

D. Comment

Too many abbreviations. Please reduce the use of abbreviations to those very nessasary.

Reply: Thank you for pointing out this. As per the suggestion, to enhance the readability we

have reduced the use of abbreviations in the paper.

E. Comment

PHQ8 and PHQ 9 differ, and their results should not be combined. Why not limiting the

results to 8 items.

Reply: We agree with the observation of the reviewer. The DAIC-WoZ Dataset is annotated for

depression using the PHQ-8 scale. Hence we have omitted any mention of PHQ-9 present in

the paper as suggested.

F. Comment

If PHQ is used, why do you discuss and cite Hamilton measure too?

Reply:

Thank you reviewer for this observation. We have mentioned Hamilton measure in the literature

survey of the paper to make the paper inclusive of readers belonging to non-medical backgrounds

too. Hence, the sole intention of mentioning Hamilton scale was to inform the user about several

techniques used by the clinicians to label mental health data.

G. Comment

We need tables that describe the participants.

Reply:

Thank you for the suggestion. We have added a table that discusses the composition of the

dataset with respect to participants according to the gender. Moreover, additional details are not

available as the Daic-Woz Dataset is provided by the AVEC 2016 Challenge and the released

baseline paper has limited information about the participants.

H. Comment

Show us correlation between all variables. (Pearson r)

Reply:

Thank you for the suggestion. As we have used a deep learning model based on tensor

factorization, the input to the method is 3D Tensors, which are stack of 2D Mel-spectrogram

tensors along the third dimension. As such Pearson r coefficient of the input tensor is not defined.

Moreover, since the stack of deep learning layers learn abstract information in the hidden layers,

which are unknown in general, it is infeasible to provide correlation of variables in this particular

proposed method.

I. Comment

Please use MDD for major depression. DOes not need the letter ”S”.

Reply: Thank you for the suggestion. We have made the suggested changes.

II. COMMENTS FROM REVIEWER 2

A. Comment

The title of the article says depression recognition so the authors are only specific parameters

based patterns generated to Recognition or authors may look the contextually fit then instead

recognition detection is more suitable.? Justify with your answer.

Reply:. We thank the reviewer for the question. As suggested , we have replaced ” detection”

with ”recognition” in line with the standards used in similar research papers such as -

1. L. Yang, D. Jiang, W. Han and H. Sahli, ”DCNN and DNN based multi-modal depression

recognition,” 2017 Seventh International Conference on Affective Computing and Intelligent

Interaction (ACII), 2017, pp. 484-489, doi: 10.1109/ACII.2017.8273643.

2.Hongying Meng, Di Huang, Heng Wang, Hongyu Yang, Mohammed AI-Shuraifi, and Yun-

hong Wang. 2013. Depression recognition based on dynamic facial and vocal expression features

using partial least square regression. In Proceedings of the 3rd ACM international workshop on

Audio/visual emotion challenge (AVEC ’13). Association for Computing Machinery, New York,

NY, USA, 21–30. https://doi.org/10.1145/2512530.2512532

B. Comment

The Word GIVE must be justified or reference adds for abbreviation if taken from research

sources. Give the significant reasons for capitalization of the word

Reply:. Thank you for pointing out this. We have removed the word ”GIVE” as it was not

adding any extra information.

C. Comment

In page 1 line no. 6 of Introduction section Major Depression Disorder(MSDD) and Multi

system depressive disorder must be given proper relation and both abbreviations cannot be

interchangeable hence look into these point and justify your comments.

Reply: We thank the reviewer for pointing out this mistake. We have made the necessary

correction and removed” Multisystem Depressive Disorder ” as this was out of context here.

D. Comment

In Line No.10 page 2 the numbers added with the symbols of percentage and again word used

so redundancy in information may be corrected. Ex: 80% percentage is not correct 80 percent

or 80% along correct. Kindly justify your action on this modification

Reply: We thank the reviewer for pointing out this mistake. We have corrected as per the

suggestion and only kept ”%” symbol and removed the word ”percent” to reduce redundant

information.

E. Comment

Line 18 of page 2 “P.H.Q” abbreviated word usability in the entire document must be con-

sistent.

Reply: We have modified the abbreviation according to the suggestion of the reviewer

F. Comment

Differentiation of the depression detection and depression recognition must be distinguished

while using the work in the article. The usability of the word must be maintained in a consistent

manner.

Reply: We thank the reviewer for this valuable comment. As per the suggestion of the reviewer,

we have maintained uniformness across the paper by replacing ”detection” with ”recognition” as

per the standards in the existing literature which utilizes deep learning in depression recognition.

G. Comment

Appreciate the authors with more grammatically correct and with simple sentence breaks made

throughout the article in a consistent manner.

Reply: We thank the reviewer for raising this concern. We have tried to simplify sentences and

improve grammar wherever possible in the paper.

H. Comment

Authors can justify properly the use of Abbreviations separated full stops and some places

without them. Entire articles use a uniform process.

Reply: We thank the reviewer for pointing out this mistake. As per the suggestion, we have

made the abbreviations uniform across the article.

I. Comment

Page No. 8 Section 3.3.2 given equations need to be numbered.

Reply: Thank you for the suggestion. We have added equation numbers for the same.

J. Comment

Page no. 9 and content in Table 1. Captions of the Row must be added to the percentage

symbols as the accuracy is measured with ratios.

Reply: We have made the necessary changes as per the reviewer’s suggestion.

K. Comment

Authors suggested rewriting the Abstract and Conclusions with simple sentences by appro-

priate breaks and conveying the authors view properly.

Reply: Thank you for the suggestion. We have tried to incorporate the reviewer’s suggestion.

L. Comment

The technical concepts are good and explanation flow is appreciable.

Reply: We thank the reviewer for his kind appreciation.

III. C OMMENTS FROM THE EDITOR

A. Comment

In the related work, need to include more recently published work with explanation of what

is the role of the deep learning research in depression analysis?

Reply:. We thank the editor for this suggestion. We have added reference to the more recent

work as well as emphasized on the importance of deep learning for mental health diagnosis.

B. Comment

I suggest to more detailed explanation for Evaluation based on framework for automated

depression classification and the compression with others recent work.

Reply:. We thank the editor for the suggestion. We have explained the evaluation procedure in

detail in section 3.4 along with details on the hyperparameter values used. Also, the comparison

with other recent work is limited to only those studies which have used accuracy and F1 score

as metric and evaluated on test partition . Most of the other recent study has used regression

metric and validation partition for evaluation, which is not included in our table of comparison.

C. Comment

Please explain in briefly and move their explanation in a paragraph explaining what each table

and figures refer to.

Reply:. We thank the editor for the suggestion. We have added details about the information

flow across various components in the diagram in the captions .

D. Comment

It will be helpful to have a few lines explaining what is inside the features chosen and why

a particular fusion method is suitable for the particular depression analysis.

Reply:. We thank the editor for this suggestion. We have added a subsection titled ” Discussion”

which briefly explains the choice of mel-spectrograms as speech feature and what information

3D TFNN based MIL architecture tries to capture in its various layers.

---

## [Decision Letter · Decision Letter 1]

25 Jul 2022

A deep tensor-based approach for automatic depression recognition from speech utterances

PONE-D-22-04732R1

Dear Dr. Pandey,

We’re pleased to inform you that your manuscript has been judged scientifically suitable for publication and will be formally accepted for publication once it meets all outstanding technical requirements.

Kind regards,

Dhananjay Singh, Ph.D.

Academic Editor

PLOS ONE

Additional Editor Comments (optional):

The overall quality of communication is good although proofreading will be needed.

Reviewers' comments:

Reviewer's Responses to Questions

**Comments to the Author**

1. If the authors have adequately addressed your comments raised in a previous round of review and you feel that this manuscript is now acceptable for publication, you may indicate that here to bypass the “Comments to the Author” section, enter your conflict of interest statement in the “Confidential to Editor” section, and submit your "Accept" recommendation.

Reviewer #3: All comments have been addressed

2. Is the manuscript technically sound, and do the data support the conclusions?

Reviewer #3: Yes

3. Has the statistical analysis been performed appropriately and rigorously? 

Reviewer #3: Yes

4. Have the authors made all data underlying the findings in their manuscript fully available?

Reviewer #3: Yes

5. Is the manuscript presented in an intelligible fashion and written in standard English?

Reviewer #3: Yes

6. Review Comments to the Author

Reviewer #3: The authors have addressed the reviewer's concerns and the revised version of the manuscript appears to be good.

7. PLOS authors have the option to publish the peer review history of their article (what does this mean?). If published, this will include your full peer review and any attached files.

Reviewer #3: No
